# Astroglial Hemichannels and Pannexons: The Hidden Link between Maternal Inflammation and Neurological Disorders

**DOI:** 10.3390/ijms22179503

**Published:** 2021-09-01

**Authors:** Juan Prieto-Villalobos, Tanhia F. Alvear, Andrés Liberona, Claudia M. Lucero, Claudio J. Martínez-Araya, Javiera Balmazabal, Carla A. Inostroza, Gigliola Ramírez, Gonzalo I. Gómez, Juan A. Orellana

**Affiliations:** 1Departamento de Neurología, Escuela de Medicina and Centro Interdisciplinario de Neurociencias, Facultad de Medicina, Pontificia Universidad Católica de Chile, Santiago 8330024, Chile; jcp.villalobos@gmail.com (J.P.-V.); tanhia.alvears@utem.cl (T.F.A.); andes.rojas.liberona@uc.cl (A.L.); cmartineza@utem.cl (C.J.M.-A.); javi.balma24@gmail.com (J.B.); c.inostrozavenegas@gmail.com (C.A.I.); gigliolaramirez@hotmail.com (G.R.); 2Institute of Biomedical Sciences, Faculty of Health Sciences, Universidad Autónoma de Chile, Santiago 8910060, Chile; claulucerom@gmail.com (C.M.L.); gonzalo.gomez@uautonoma.cl (G.I.G.)

**Keywords:** connexins, pannexins, hemichannels, pannexons, neuroinflammation, lipopolysaccharide, neuron, astrocyte, microglia, neurodegeneration, excitotoxicity

## Abstract

Maternal inflammation during pregnancy causes later-in-life alterations of the offspring’s brain structure and function. These abnormalities increase the risk of developing several psychiatric and neurological disorders, including schizophrenia, intellectual disability, bipolar disorder, autism spectrum disorder, microcephaly, and cerebral palsy. Here, we discuss how astrocytes might contribute to postnatal brain dysfunction following maternal inflammation, focusing on the signaling mediated by two families of plasma membrane channels: hemi-channels and pannexons. [Ca^2+^]_i_ imbalance linked to the opening of astrocytic hemichannels and pannexons could disturb essential functions that sustain astrocytic survival and astrocyte-to-neuron support, including energy and redox homeostasis, uptake of K^+^ and glutamate, and the delivery of neurotrophic factors and energy-rich metabolites. Both phenomena could make neurons more susceptible to the harmful effect of prenatal inflammation and the experience of a second immune challenge during adulthood. On the other hand, maternal inflammation could cause excitotoxicity by producing the release of high amounts of gliotransmitters via astrocytic hemichannels/pannexons, eliciting further neuronal damage. Understanding how hemichannels and pannexons participate in maternal inflammation-induced brain abnormalities could be critical for developing pharmacological therapies against neurological disorders observed in the offspring.

## 1. Introduction

Clinical evidence has established that environmental clues acting at specific windows during fetal development affect lifelong trajectories across health and disease [1]. Such “programming” encompasses a physiological and adaptive process that sculpts the structure and function of different tissues at the stage when they are most plastic due to the proliferation and differentiation of progenitor cells [2]. Nevertheless, negative gene-environment interactions linked to perinatal disease, either maternal or fetal, disrupt this physiological programming, which increases individual susceptibility to develop complex diseases from birth to adult life [3]. For instance, maternal immune perturbations during pregnancy, either in response to infections or noninfectious stimuli (e.g., diabetes, stress, maternal allergic asthma, obesity, or toxin exposures), cause enduring or later-in-life alterations of the offspring brain structure and function [4]. The latter increases the risk of developing several psychiatric and neurological disorders, including schizophrenia, intellectual disability, bipolar disorder, autism spectrum disorder (ASD), microcephaly, and cerebral palsy [5] (Figure 1).

Although the study of Karl A. Menninger and later on the work of Torrey and Peterson were pioneering in revealing the association between viral infection and subsequent psychotic disease [6,7], it was Mednick et al. (1988) who showed that maternal influenza enhances the incidence of schizophrenia in the offspring [8]. Thenceforth, similar findings have been observed with other viral (e.g., cytomegalovirus, herpes simplex virus type 2, varicella-zoster and polio), bacterial (e.g., sinusitis, tonsillitis, pneumonia, and pyelonephritis), and parasite (e.g., toxoplasmosis) infections [9,10]. In the same line, other studies have connected rubella and cytomegalovirus infection during pregnancy with increased risk of ASD in the offspring [11,12], whereas cerebral palsy in adulthood associates with maternal infections [13,14]. Although less well understood and studied, intellectual disability and bipolar disorder correlate with bacterial and *Toxoplasma gondii* infection, respectively, during pregnancy [15,16]. Recent studies have hypothesized that prenatal exposure to SARS-CoV-2, the virus that causes the coronavirus disease 2019 (COVID-19), could augment the incidence of psychosis, schizophrenia, and schizophrenia spectrum disorders in the offspring [17]. 

Although epidemiological research provides a strong link between prenatal life and adult neurological disease risk, its efficiency in deciphering the concomitant downstream cellular and molecular mechanisms is limited for ethical or technical reasons. Thus, the vast amount of knowledge gathered to date about maternal inflammation that results in offspring brain abnormalities comes from experimental studies in rodents (for a comprehensive review, see [18]). Most of them have used the systemic administration of lipopolysaccharide (LPS) or polyriboinosinic-polyribocytidilic acid [poly (I:C)] during pregnancy [18]. LPS, the major component of the outer membrane of Gram-negative bacteria, is a well-known, established bacterial infection model, which leads principally to cytokine production, inflammation, fever, complement cascade activation, hypothalamic–pituitary–adrenal axis activation, and sickness behavior [19]. At the other end, poly (I:C) is a synthetic analog of double-stranded RNA that efficiently mimics the acute phase response to viral infection, including the production and release of interleukin (IL)-1β, IL-6, and tumor necrosis factor (TNF)-α, as well as the induction of the type I interferons (IFNs): IFN-α and IFN-β [20]. Several studies using these immunogenic approaches have demonstrated that maternal inflammation impairs normal behavior and social interactions in adult progeny [21,22,23,24,25] (Figure 1). The latter includes a decline in learning and memory, increased anxiety-like and repetitive behaviors, motor deficits, and disturbed exploratory performance [21,25,26,27,28,29]. 

There is a certain consensus that a common pathogenic pathway linked to cytokine-mediated inflammation disrupts fetal brain development and adult central nervous system (CNS) maturation following maternal disease [4,30,31]. Coherent with this notion, human epidemiological evidence indicates that high gestational levels of IL-1α, IL-6, IL-8, IFN-γ, TNF-α, granulocyte macrophage colony-stimulating factor, and C-reactive protein augment the incidence of schizophrenia and ASD in the progeny [32,33,34] (Figure 1). This evidence harmonizes with the critical role of IL-1β and IL-17 A on adult brain abnormalities observed following maternal immune activation in rodents [35,36]. In fact, in the absence of a pathogenic agent, the administration of IL-6 during pregnancy is sufficient to promote multiple behavioral and cognitive abnormalities in the offspring [37]. Despite the significant similarities between the inflammatory responses induced by several models of maternal inflammation [38,39,40], they also have differential immune signatures and specific pathophysiological responses that impact brain development, structure, and function [4,18,26]. For instance, unlike LPS, poly (I:C) is a potent activator of type I IFNs (e.g., IFN-β) and consequent antiviral immune responses [20], whereas LPS is more proficient in inducing the production and release of TNF-α from macrophages [41]. This predilection of LPS for TNF-α instead type I IFN signaling could explain why this endotoxin is more robust than poly (I:C) in provoking anorexia, lethargy, and fever [42].

The myriad of inflammatory factors produced by infections or noninfectious pathological stimuli during gestation induces diverse pathophysiological processes in the maternal, placental, and fetal compartments [4]. Part of these mediators can cross the blood-placental barrier, triggering systemic fetal inflammation and oxidative stress [43] and affecting the brain, with potentially damaging consequences for neuronal and glial cell function, synaptic transmission and plasticity, and behavior [44]. In the CNS, the innate immune system integrates these complex immune responses, whose central member is the microglia. These cells are the first line of defense against internal or external agents that resist or resolve harmful threats to restore homeostasis [45]. The role of microglia in fetal programming and postnatal brain abnormalities has been extensively studied [46,47,48,49,50]; however, the implication of other crucial glial populations involved in neuroinflammation remains elusive: the astrocyte [51]. Here, we discuss how astrocytes might contribute to postnatal brain dysfunction following maternal inflammation, focusing on the signaling pathways mediated by two families of plasma membrane channels: hemichannels and pannexons.

**Figure 1 ijms-22-09503-f001:**
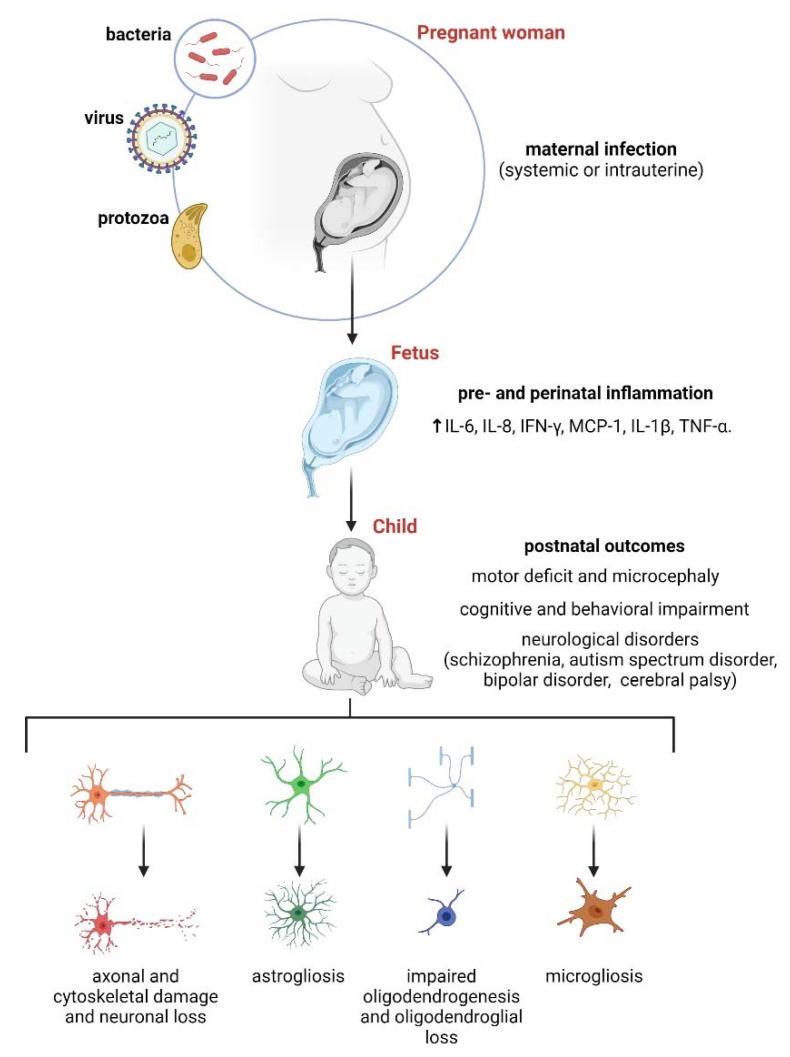
Schematics showing general aspects of maternal infection and its contribution to offspring brain abnormalities. Diverse infectious agents such as viruses (e.g., influenza), bacteria (e.g., *E. coli*), or protozoa (e.g., *Toxoplasma gondii*) can induce systemic or intrauterine maternal inflammation. The latter occurs in parallel with the activation of placental cytokine receptors and immune cell infiltration, with both phenomena being crucial for pre and perinatal fetal brain inflammation. Cytokine-mediated inflammation at this stage causes severe consequences for fetal brain development and potentially elicits diverse postnatal CNS alterations, including neuronal damage [52,53], reactive astrogliosis and microgliosis [48,51], and impaired oligodendrogenesis and oligodendroglial loss [54,55]. Certainly, fetal brain injury caused by the pathogens above increases the postnatal risk of developing motor deficits, microcephaly, and cognitive and behavioral impairment. In addition, multiple neurological disorders associate with prenatal inflammation: schizophrenia, autism spectrum disorder, bipolar disorder, and cerebral palsy.

## 2. Astrocytes: Emerging Stars in the Healthy and Diseased Brain

Mounting evidence in the last decades has refuted the notion that astrocytes act as simple fostering and buffering elements in the CNS [56]. Intracellular Ca^2+^ ([Ca^2+^]_i_) waves within and among astrocytes encompass a time-scale mechanism for allowing rapid intra- and inter-cellular signaling at different hierarchies [57,58,59,60]. These processes begin with the extracellular influx of Ca^2+^ via ion channels and through Ca^2+^ release from intracellular stores, causing [Ca^2+^]_i_ transients that vary in frequency, kinetic and spatial spread according to the astrocyte anatomical zone [61]. Braced with this equipment and in the companion of pre- and postsynaptic neurons, astrocytes constitute the “tripartite synapse”—the angular stone of the chemical synaptic transmission—where they monitor neurotransmission and react to it by the [Ca^2+^]_i_-dependent release of signals that control neuronal activity termed “gliotransmitters” (i.e., glutamate, D-serine, and ATP) [62]. Accordingly, astrocytes seem crucial for synaptic transmission and plasticity, and learning and memory consolidation [63]. 

Furthermore, during high neuronal activity, astrocytes produce [Ca^2+^]_i_ signals that spread locally in networks in the form of [Ca^2+^]_i_ waves that reach specialized astrocytic terminal processes or “endfeet” that contact the vasculature [64]. There, vasoactive messengers are released, allowing astrocytes to regulate the cerebral blood flow and exchange of energy-rich metabolites, with potentially significant consequences for neuronal firing, synaptic plasticity and higher brain functions [65]. In particular, astrocytic endfeet takes up glucose and distributes it among astrocytes through intercellular connections termed gap junctions [66]. Depending on neuronal energy demand, glucose can be stored in the form of glycogen or used by the astrocyte in glycolysis, being the resulting pyruvate converted to lactate and then released into the extracellular space [67]. Then, neurons can take up this lactate, convert it into pyruvate, and utilize it in aerobic respiration within the mitochondria [68] or take up directly glucose from the interstitial space and generate ATP from glycolysis and oxidative metabolism [69]. In addition, astrocytes participate in the innate immune response and govern the homeostasis of the brain interstitial fluid, supplying neurons with precursors for biosynthesis, controlling pH and K^+^ homeostasis and recycling glutamate, oxidized scavengers, and other waste products [70]. 

Given that neurons are particularly susceptible to the action of free radicals and reactive oxygen species (ROS) [71,72], astrocytes protect them from this not only by providing an extensive array of antioxidant molecules and ROS-detoxifying enzymes (e.g., glutathione, superoxide dismutase, and glutathione peroxidase) but also through the direct transference of their healthy mitochondria, as well as the degradation of axonal mitochondria [73,74]. Astrocytes require an efficient [Ca^2+^]_i_ regulation mechanism to fulfill their synaptic, metabolic, and homeostatic roles [60]. In this matter, the function of astrocyte mitochondria seems to be pivotal [75]. Along with serving as a source of ATP to fuel several Ca^2+^ pumps that keep low the [Ca^2+^]_i_, mitochondria can also actively import Ca^2+^. The membrane potential associated with the proton electrochemical gradient across the inner mitochondrial membrane (~180–200 mV) facilitates the import of Ca^2+^ into the mitochondria against its concentration gradient via a Ca^2+^ uniporter protein [76]. Moreover, mitochondria can release Ca^2+^ from its matrix towards the cytosol via the mitochondrial Na^+^/Ca^2+^ exchanger [77] and the opening of the mitochondrial permeability transition pore [78]. Mitochondria serve as both sink and source of Ca^2+^ in astrocytes, thereby regulating the frequency, amplitude, and half-life of Ca^2+^ transient events in their cytoplasmic processes with significant potential consequences for astroglial signaling and function [75,79,80]. 

Under pathological conditions, astrocytes experience a long-lasting morphological, molecular, and functional change referred to as “reactive astrogliosis”, which is characterized by cytoskeletal rearrangements, hypertrophy, increased expression of the glial fibrillary acidic protein (GFAP), loss of structural complexity, metabolic alterations, and release of inflammatory mediators [81]. While this process is an adaptive mechanism necessary for limiting acute injury and favoring wound repair, when persistent, it can turn into a detrimental response if astrocytes neglect their supportive role toward neurons [82]. Reactive astrogliosis becomes dysfunctional when damage is intense and chronic and usually negatively impacts different astrocyte aspects such as gliotransmission, Ca^2+^ signaling, mitochondrial function, antioxidant defense, and inflammatory response and survival [83]. 

## 3. Maternal Inflammation and Its Impact on Astrocytes

Clinical and animal studies have revealed that maternal inflammation causes different morphological and functional alterations on astrocytes. For instance, necrosis observed in periventricular leukomalacia, a brain injury that induces cerebral palsy, likely via maternal infection [84], correlates with increased GFAP and IFN-γ expression in astrocytes, as well as nitrosative and oxidative damage [85,86]. Consistent with this, maternal LPS administration accentuates offspring astrogliosis in the hippocampus, cortex, amygdala, hypothalamus, thalamus, and white matter, associated with hypomyelination [87,88,89,90,91]. Likewise, long-lasting astrogliosis takes place in the hippocampus of mice prenatally exposed to the human influenza virus [92], and similar findings have been observed following poly (I:C)-induced maternal immune activation [93,94,95,96]. Despite the latter, other studies have described that prenatal poly (I:C) exposure does not affect the astroglial number and expression of GFAP in the offspring [97,98,99,100]. These apparent discrepancies between the studies mentioned above make sense in the light of at least two crucial factors. Firstly, astrogliosis is biologically complex and cannot be reduced to the expression of one marker, which is significant, considering that GFAP^+^ astrocytes represent just a fraction of the total astroglial population with a substantial regional (and probably developmental) heterogeneity [101]. In addition, counting GFAP^+^ astrocytes illustrates presumably GFAP expression alterations rather than valid changes in astrocyte number. At the other end, the experimental design of poly (I:C)-induced maternal inflammation differs considerably among studies, with different doses, administration routes, and gestational time points of exposure. These factors undoubtedly change the fetal nıche to different degrees, affecting the outcome of astroglial function and reactivity significantly.

Alterations to morphology and number of astrocytes occur in the offspring of animals exposed to other stressors or challenges during pregnancy, including (but not exclusively) IL-6 [53], perfluorooctane sulfonate [102], stress [103,104], ischemia [105], dexamethasone [106], ethanol [107], carbon black nanoparticles [108], and high-fat diet [109,110]. Some studies have shed light on possible mechanisms explaining how astrocytes may contribute to prenatal life-induced programming of the brain. For example, Zhang and colleagues demonstrated that prenatal LPS exposure produces prolonged glutamate elevation in periventricular white matter in the progeny associated with astroglial hypertrophy and decreased glial L-glutamate transporter 1 [46]. On the other hand, prenatal stress increases astroglial death and GFAP expression, accompanied by elevated production of fractalkine and nitric oxide (NO) [104]. Both studies harmonize with previous data indicating that glutamate and NO released by astrocytes impair neuronal function and survival [111,112,113]. More recently, two studies have suggested that maternal inflammation-induced brain abnormalities in adulthood depend on the persistent activation of two families of large-pore plasma membrane channels in astrocytes: hemichannels and pannexons. 

## 4. Hemichannel and Pannexons: Protagonists on Astroglial Physiology and Pathophysiology

In the last decade, other research groups and we have described that hemichannels and pannexons, two families of plasma membrane channels, may alter different aspects of astroglial function with potentially significant consequences for neuronal function during pathological conditions [111,112,114,115,116,117,118,119,120,121,122]. Hemichannels result from the oligomerization of six protein subunits called connexins around a central pore [123] (Figure 2). Connexins encompass a highly conserved protein family encoded by 21 genes in humans and 20 in mice, with orthologs in other vertebrate species [124]. For a considerable time, the essential function ascribed to hemichannels was to constitute the basic components of the gap junctions, these being aggregates of intercellular channels that provide the direct but selective molecular and ionic exchange between the cytoplasm of contacting cells [125] (Figure 2). Notwithstanding, in the 90 s, groundbreaking findings by Paul and colleagues revealed the presence of functional and solitary hemichannels in “non-junctional” membranes [126]. Nowadays, it is well-established that these channels act like permeable pores, providing a diffusional route for the release of relevant quantities of autocrine and paracrine signaling molecules (e.g., ATP, glutamate, D-serine, NAD^+^, and PGE_2_) as well as the influx of other substances (i.e., Ca^2+^, cADPR, and glucose) [123] (Figure 2). 

Two decades ago, a novel family of three membrane proteins called pannexins (Panxs 1–3) was discovered, with the ability to constitute single membrane channels (also known as pannexons) that connect the cytosol with the interstitial space [127]. Even though hemichannels and pannexons belong to a broad family of large-pore channels [128], they diverge in terms of permeability and conductance, as well as gating and posttranslational mechanisms that modulate them [129,130]. Consistent with this idea, both channels remain fully functional under resting membrane potentials [131,132], displaying large membrane currents after depolarization [126,131]. Unlike hemichannels, pannexons produce peak current amplitudes with fast kinetics and show larger unitary conductance, weak voltage-gating, and diverse subconductance states [126,131,133,134,135,136]. Similarly, hemichannel activity is significantly modulated by the extracellular concentration of divalent cations [137], whereas gating properties of pannexons remain insensitive to external Ca^2+^ [133,138].

Connexin 43 (Cx43) constitutes the most ubiquitous connexin expressed by astrocytes [139]. Astrocytes also exhibit appropriate levels of Panx1, and both Cx43 and Panx1 form functional astroglial hemichannels and pannexons, respectively, on in vitro and ex vivo preparations [118,140,141,142,143]. Cellular signaling and gliotransmitter release via the opening of astrocytic hemichannels and pannexons underpin relevant biological processes at the nervous system, including neuronal oscillations [144], astroglial migration [145], food intake [146], synaptic transmission, and plasticity [132,147,148,149], as well as memory consolidation and behavior [150,151] (Figure 3). Despite the above, the uncontrolled activation of these channels in astrocytes associate with the pathogenesis and progression of homeostatic imbalance in various neuropathological diseases [111,112,114,115,116,117,118,119,120,121,122]. The nature of the pathological agents linked to the opening of astroglial hemichannels/pannexons is multiple, including cytokines [152], amyloid-β-peptide [111], α-synuclein [121], ethanol [122], anticonvulsants drugs [153], ultrafine carbon black particles [154], and oxidant stress [155]. At least three mechanisms have linked the persistent opening of hemichannels and pannexons with cell dysfunction and damage. At one end, the uncontrolled entry of Na^+^ and Cl^−^ through hemichannels may result in osmotic and ionic imbalances linked to further cell swelling and plasma membrane breakdown [126,156] (Figure 3). Relevantly, hemichannels are permeable to Ca^2+^ [157,158], which could allow its influx to the cytosol during pathological conditions. In the same line, Panx1 channels release ATP to the interstitial space, which activates purinergic receptors, causing the entry of extracellular Ca^2+^ or its release from intracellular stores [159]. The direct or indirect increase in [Ca^2+^]_i_ mediated by hemichannels/pannexons could lead to Ca^2+^ overload and consequent induction of different proteases, phospholipases, and other hydrolytic enzymes, as well as oxidative stress and caspase activation [119,160]. Last but not least, exacerbated hemichannel/pannexon activity may trigger the release of high amounts of molecules potentially toxic for neighboring cells, such as glutamate, in the case of the CNS [111,112] (Figure 3).

## 5. Connecting Maternal Inflammation with the Activation of Hemichannels and Pannexons in Offspring Astrocytes 

Avendaño and collaborators were pioneers in demonstrating that LPS administration during pregnancy augments the activity of Cx43 hemichannels and Panx1 channels in neonatal astrocytes cultures [170]. How could LPS trigger the opening of these channels in the context of fetal programming? As mentioned above, maternal administration of LPS causes an acute phase of systemic inflammation that goes hand in hand with the production of placental inflammatory mediators, which occurs at crucial developmental stages of the fetal brain [43]. Although LPS induces the activation of Cx43 hemichannels in cultured astrocytes and C6 glioma cells [152,171], the ability of prenatal LPS exposure to promote the opening of these channels likely relies on downstream interactions of cytokines with placental receptors, which significantly and permanently affect the structure and functional capacity of the fetal brain. Accordingly, prenatal LPS-induced activation of Cx43 hemichannels and Panx1 channels is mitigated by the inhibition of IL-1β/TNF-α signaling and accompanied by high autocrine production of astroglial IL-1β/TNF-α [170]. The latter agrees with the fact that both cytokines directly: (i) enhance ion currents mediated by astrocytic Cx43 hemichannels [152] and (ii) modulate the uptake of cationic molecules via these channels [172]. Relevantly, the cytokine-mediated cell influx of small molecules is dependent on the properties of the permeant species (e.g., ethidium, 2-NBDG, DAPI) [172]. Altogether this evidence suggests that maternal exposure to LPS modifies the in utero fetal brain environment, and thus, it shapes the function of astroglial hemichannels and pannexons in the offspring. Although certainly connexins are controlled by epigenetics [173], the involvement of DNA methylation, histone acetylation, or microRNA regulation in the above phenomenon remains largely ignored. 

Is there another source, besides astrocytes, for the production of IL-1β/TNF-α following maternal inflammation? Recently, it was revealed that microglia via the above cytokines elicit the prenatal LPS-mediated activation of astrocyte Cx43 hemichannels and Panx1 channels [174]. This evidence comes from experiments showing the ameliorative effect of minocycline, a molecule that mitigates microglial activation, or inhibition of IL-1β/TNF-α signaling, in the long-lasting opening of these channels in offspring hippocampal astrocytes [174]. These data are coherent with previous studies revealing that the LPS-mediated release of IL-1β and TNF-α from microglia promotes the activation of astroglial Cx43 hemichannels in vitro and ex vivo [114,152]. IL-1β/TNF-α signaling in astrocytes activates p38 mitogen-activated protein kinase (p38 MAPK), resulting in the expression of the inducible NO synthase (iNOS) and further NO production [175,176]. In agreement with this, blockade of p38 MAPK or iNOS strongly prevents the enhanced astrocyte hemichannel/pannexon activity observed in neonatal cultures or adult brain slices from prenatally LPS-exposed offspring [170,174]. The NO-mediated S-nitrosylation of Cx43, a posttranslational modification that opens Cx43 hemichannels [155], might play a fundamental role in this phenomenon as iNOS expression and production of NO show increments in astrocytes from the offspring of LPS-exposed dams [170,174] (Figure 4). This evidence also agrees with the higher production of IL-1β, TNF-α, and NO found in the brain of prenatally LPS-exposed offspring [18,177,178]. 

Given that NO reduces the opening of Panx1 channels either by S-nitrosylation [179] or PKG-dependent phosphorylation [180], their activation evoked by prenatal LPS possibly materialize due to alternative mechanisms. A significant component in opening Panx1 channels arises from ATP signaling and subsequent activation of purinergic receptors [181]. Panx1 co-immunoprecipitates with P2X_7_ receptors (P2X_7_Rs) [182,183] and seems to establish protein-to-protein interactions with them through the proline 451 in the C-terminal tail of P2X_7_R [184,185]. Interestingly, prenatal LPS exposure induces the release of ATP from astrocytes in vitro and ex vivo by a mechanism involving the activation of both Panx1 channels and P2X_7_ receptors [170,174] (Figure 4). This result is consistent with other findings showing that ATP induces its release through a positive loop that implicates the opening of Panx1 channels and subsequent activation of P2X_7_Rs [117,141]. As in the case of IL-1β and TNF-α, both microglia and/or astrocytes could act as sources for ATP signaling in the prenatally LPS-exposed adult offspring, thereby activating distant glial cells via P2X_7_Rs. If so, the activation of P2X_7_Rs switches off due to the decrease of ATP by its diffusion to distant interstitial areas as its degradation by extracellular exonucleases. Alternatively, ATP could impede its release by directly blocking Panx1 channels [186]. Intriguingly, prenatal LPS enhances the expression of NLRP3 inflammasome and levels of IL-1β [187], while P2X_7_R-mediated opening of Panx1 channels causes IL-1β secretion via activation of the inflammasome in different cell types, including astrocytes [182,188,189,190,191]. Whether or not the inflammasome contributes to the opening of astrocyte Panx1 channels following maternal LPS exposure has not been clarified and requires further study.

Is it Ca^2+^ signaling involved in prenatal LPS-induced hemichannel/pannexon opening in astrocytes? Both Cx43 hemichannels and Panx1 channels augment their activity upon a moderate rise in [Ca^2+^]_i_ [147,159,192,193]. Of note, astrocytes from prenatally LPS-exposed offspring display increased spontaneous [Ca^2+^]_i_ oscillations with large amplitude, which was found decisive for the activation of Cx43 hemichannels in these cells [170,174] (Figure 4). More critical, this response causes the Cx43 hemichannel-dependent release of glutamate and subsequent rise of basal [Ca^2+^]_i_ via intracellular stores, a response being underpinned by activation of metabotropic glutamate receptor subtype 5 (mGluR5) and further downstream action of PLC and IP_3_ receptors [174]. These findings harmonize with the increased release of glutamate observed in the hippocampus of prenatally LPS-exposed offspring [194] and with the fact that mGluR5 controls [Ca^2+^]_i_ responses in astrocytes [195]. 

## 6. Repercussions of Hemichannel and Pannexon Activation in the Offspring Brain following Maternal Inflammation 

During pathological conditions, long-lasting activation of hemichannels and pannexons alters multiple aspects of astroglial function such as [Ca^2+^]_i_ homeostasis, gliotransmission, inflammasome activation, cytokine secretion, redox potential, mitochondrial dynamics and survival [114,119,120,121,163,191,196,197,198]. How might these channels contribute to maternal LPS-induced astroglial dysfunction in the offspring? The inflammatory profile and homeostatic function of astrocytes require a delicate regulation of diverse [Ca^2+^]_i_ parameters, such as frequency, amplitude, the half-life of Ca^2+^ transient events, and relaxation states [60,75,199]. Because hemichannels and pannexons, directly or indirectly, cause the influx of Ca^2+^ [157,158,200,201,202] and their opening is modulated by [Ca^2+^]_i_ [147,159,192,193], they could significantly impact the function, reactivity, and fate of astrocytes. Supporting this line of thought, in vivo postnatal administration of TAT-gap19, a specific Cx43 hemichannel blocker that crosses the blood-brain barrier [203], prevents the prenatal LPS-evoked branch arborization and hypertrophy exhibited by adult hippocampal astrocytes [174], a well-recognized feature of reactive astrogliosis [82]. Similarly, the rise in GFAP expression observed in the hippocampus of prenatally-LPS exposed adult offspring was suppressed by the administration of TAT-gap19 [174]. These findings are consistent with other studies demonstrating that TAT-gap19 decreases reactive astrogliosis and microgliosis, as well as inflammatory cytokine levels in models of intracerebral hemorrhage injury and midbrain dopamine neurodegeneration [168,204]. Of note, specific blockade of Cx43 hemichannels augments the Yes-associated protein nuclear translocation, resulting in subsequent inhibition of TLR4-NFκB and JAK2-STAT3 pathways [168], the latter being crucial reactive astrogliosis [82].

Although it is unknown whether maternal inflammation affects the survival of astrocytes, the potential impact of hemichannels/pannexons in this phenomenon deserves analysis and might occur through different mechanisms. The hemichannel/pannexon-mediated [Ca^2+^]_i_ overload could produce free radicals, lipid peroxidation, and plasma membrane damage [205]. Cytosolic Ca^2+^ might also translocate into the mitochondrial matrix, where it triggers the collapse of mitochondrial membrane potential, causing not only loss of ATP production and generation of reactive oxygen species (ROS) but also cell death via the release of cytochrome C through the mitochondrial transition pore and activation of caspase-3 [206,207]. In addition, multiple lines of work indicate that osmotic and ionic imbalances evoked by the increased influx of Na^+^ and Cl^−^ via hemichannels or pannexons could lead to subsequent cell swelling and plasma membrane breakdown [121,126,156,208]. Ultimately, as mentioned before, ATP released through Panx1 channels could be pivotal for the activation of the inflammasome, resulting in the secretion of mature IL-1β and IL-18, and the induction of pyroptosis, a lytic cell death accompanied by rapid cell-membrane rupture [188,191,209,210]. How could the transient activation of hemichannels/pannexons have a long-lasting effect on astroglial function? A possible explanation could be based on the hemichannel/pannexon-mediated [Ca^2+^]_i_ imbalance and subsequent activation of astrocytes. Ca^2+^ is known to regulate the function of diverse transcription factor pathways [211]; most of them (e.g., NFκB, JAK/STAT, FOX proteins, peroxisome proliferator-activated receptors, and activator protein-1) have been involved in sculpting gene-expression programs implicated in astrocyte activation [212]. With this in mind, once astrocytic [Ca^2+^]_i_ imbalance occurs after the activation of hemichannels/pannexons, there are various pathways through which cytosolic Ca^2+^ could sustain the reactive phenotype of astrocytes. 

Maternal inflammation impairs hippocampal-mediated cognitive behavior [21,22,24,28] and long-term potentiation [213]. Both phenomena associate with dendritic retraction of pyramidal neurons and loss of synapses in diverse neurological conditions [214,215], including maternal LPS exposure [216]. Notably, blockade of Cx43 hemichannels with TAT-gap19 completely prevents the prenatal LPS-induced reduction of hippocampal neurite arborization and length, as well as the decline in dendritic spine density [174]. Most significantly, the increased death of CA1 pyramidal neurons observed in offspring hippocampus was completely prevented by inhibiting Cx43 hemichannels in this pathological model. This evidence indicates that by altering the functions of astrocytes and/or releasing excitotoxic amounts of gliotransmitters, Cx43 hemichannels would be crucial protagonists in neuronal damage and synaptic dysfunction induced by maternal inflammation. The latter idea is strengthened in light of other antecedents showing that LPS-induced impairment of excitatory synaptic activity depends on the opening of astroglial Cx43 hemichannels [114]. 

How might prenatal LPS-induced opening of astrocytic hemichannels and pannexons impair neuronal function and survival? At one end, it is plausible to hypothesize that [Ca^2+^]_i_ imbalance linked to the opening of astrocytic hemichannels and pannexons could disturb essential functions that sustain not only astrocytic survival but also astrocyte-to-neuron support, including energy and redox homeostasis, uptake of K^+^ and glutamate, and the delivery of neurotrophic factors and energy-rich metabolites. Both phenomena, either the dysfunction of astrocytes or a reduction in their number triggered by maternal inflammation, could make neurons more susceptible to the deleterious effect of prenatal LPS exposure itself and/or the experience of a second immune challenge during adulthood (see the “two-hit” explanation for the schizophrenia etiology [217]). At the other end, it is possible to conjecture that prenatal LPS exposure could cause excitotoxicity by producing the release of high amounts of gliotransmitters via hemichannels/pannexons, eliciting further neuronal damage. In agreement with the latter idea, prenatal LPS exposure prompts the Cx43 hemichannel/Panx1 channel-dependent release of glutamate and ATP from astrocytes, making neurons in cultures or brain slices toxic [170,174] (Figure 5). Even more critical, the use of astroglial conditioned media revealed that ATP and further activation of neuronal Panx1 channels contribute to neuronal loss caused by prenatal LPS exposure [170]. 

Neurons express functional Panx1 channels [218,219], whereas their ability to constitute connexons or hemichannels is still a matter of investigation [220,221]. In other systems, it has been proposed that ATP could trigger the opening of neuronal Panx1 channels via the above-mentioned protein–protein interactions between these channels and P2X_7_Rs or via stimulation of P2Y receptors and further raising of [Ca^2+^]_i_ [111,112]. In the case of maternal inflammation, it seems that P2X_7_Rs rather than P2Y_1_Rs contribute to the astroglial ATP-mediated neuronal death in prenatally LPS-exposed offspring [170] (Figure 5). Alternatively, Panx1 channels could be opened by a rise in [Ca^2+^]_i_ and further phosphorylation of the Panx1 amino acid residue S394 by activated CaMKII, as recently demonstrated in cells subjected to membrane stretch [193]. Although glutamate released via astroglial Cx43 hemichannels increases levels of astroglial basal [Ca^2+^]_i_ following maternal inflammation [174], it remains unknown whether this gliotransmitter influences neuronal survival. If so, the activation of neuronal Panx1 channels might arise as a possible mechanism in the downstream signaling of glutamate-induced neuronal loss linked to *N*-methyl-d-aspartate receptors (NMDARs) and Src family kinase (SFK) [222]. In this multiprotein complex, the metabotropic activation of NMDARs recruits SFK to open Panx1 channels via phosphorylation of Panx1 C-terminus, producing sustained neuronal depolarizations and consequent excitotoxicity during anoxia/ischemia [222,223,224]. 

## 7. Conclusions

The CNS needs protection from endogenous and exogenous threats. The notion of the brain being a privileged organ with a poor immune capacity does not conciliate with recent evidence indicating that it performs complex immune responses primarily based on its innate immune system, a “first line” of defense ensuring brain homeostasis [225,226,227]. Along with microglia, astrocytes are cornerstones in this process as they restrain infection and eliminate pathogens, cell debris, and misfolded proteins. Astrocytes also sense neuronal activity and respond locally to it through the release of gliotransmitters that further modulate synaptic function and transmission. The findings discussed in this review support the idea that activation of astrocyte Cx43 hemichannels and Panx1 channels could contribute to offspring brain abnormalities observed following maternal inflammation. In particular, the opening of these channels could be the hidden link between brain innate immune activation occurring at early phases of fetal development and postnatal decline in synaptic function and transmission. Further studies will clarify whether astroglial hemichannel/pannexon opening evoked by prenatal inflammation takes place just at postnatal stages or during fetal development as well.

An aspect that remains puzzling is whether maternal inflammation activates astroglial hemichannels/pannexons in manners that differ in intensity and temporal kinetics depending on the nature of the immune stimuli (e.g., viruses, bacteria, and protozoa). In such a case, the outcomes in synaptic function and neuronal survival could be considerably different. It seems clear that prenatal inflammation plays a central role in opening these channels as it occurs following other maternal stressors not necessarily linked to infections. For instance, the combination of prenatal nicotine exposure and postnatal high-fat/cholesterol diet produces the activation of hemichannels/pannexons in astrocytes, microglia, and neurons, a response associated with cytokine production that does not occur when animals are exposed to these stressors separately [228]. In the same direction, maternal exposure to high doses of dexamethasone activates the NLRP3 inflammasome, which results in the opening of oligodendrocyte hemichannels in a P2X_7_R-dependent manner [229]. Understanding how hemichannels and pannexons participate in the impairment of astrocyte-neuron crosstalk during and after maternal inflammation could be critical for developing pharmacological therapies against neurological disorders observed in the offspring.

## Figures and Tables

**Figure 2 ijms-22-09503-f002:**
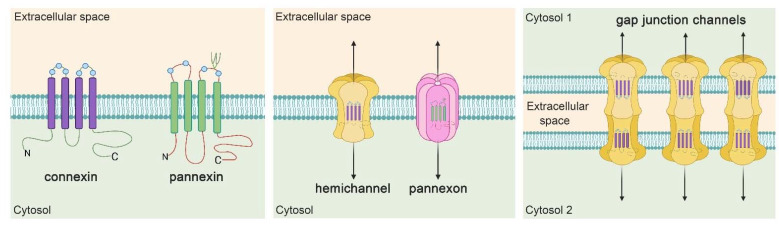
The general structure of connexin- and pannexin-based channels. Connexins and pannexins share a similar membrane topology with four α-helical transmembrane domains connected by two extracellular loops and one cytoplasmic loop; both the amino- and carboxy-termini are intracellular (left panel). The relative positions of the extracellular loop cysteines (blue light balls) and glycosylated asparagine (green branches) are also showed. Hemichannels (also known as connexons) are formed by the oligomerization of six subunit connexins around a central pore, whereas pannexons are constituted of seven pannexin subunits (middle panel). Both channels underpin the ionic and molecular interchange between the intra- and extracellular milieu. In addition, hemichannels dock each other to build intercellular channels termed gap junction channels (right panel). These channels aggregate in anatomical structures called gap junctions to support the intercellular cytoplasmic exchange of metabolites, second messengers, and ions.

**Figure 3 ijms-22-09503-f003:**
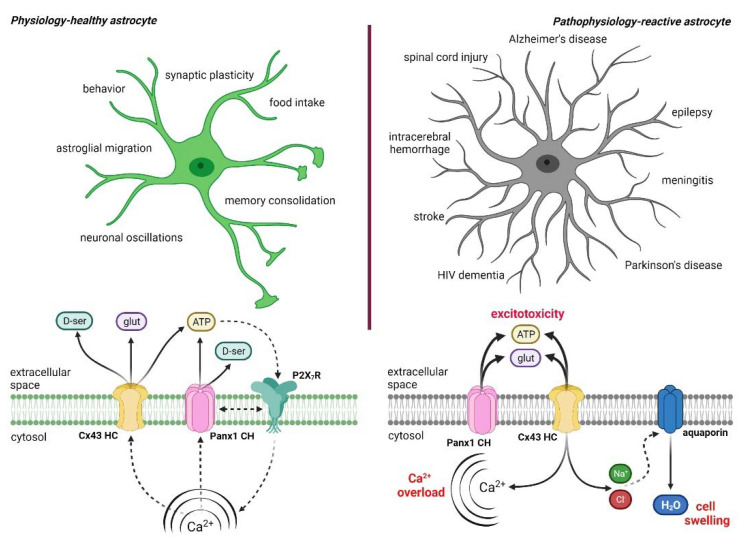
Physiological and pathophysiological roles of astrocytic Cx43 hemichannels and Panx1 channels in the CNS. The top left panel represents a healthy astrocyte (in green) under physiological conditions. In this context, the physiological release of gliotransmitters (glutamate, D-serine, and ATP) via astrocytic Cx43 hemichannels and Panx1 channels, along with [Ca^2+^]_i_ and purinergic receptor signaling, contributes to diverse biological brain processes (bottom left panel), including synaptic plasticity [132,147,148,149], neuronal oscillations [144], food intake [146], astroglial migration [145], fear memory consolidation [151], and behavior [150]. The top right panel shows a reactive astrocyte (in gray) in a pathophysiological scenario. Here, reactive astrogliosis depicted by hypertrophy of cellular processes is accompanied by persistent and exacerbated opening of Cx43 hemichannels and Panx1 channels in astrocytes (bottom right panel). The latter likely leads to cellular damage and dysfunction by different mechanisms. For example, Ca^2+^ influx via Cx43 hemichannels might activate phospholipase A_2_ and subsequently elicit the production of arachidonic acid and stimulation of the cyclooxygenase/lipoxygenase pathway. This response could then increase levels of free radicals, lipid peroxidation, and plasma membrane damage. In addition, Na^+^ and Cl^−^ entry via Cx43 hemichannels/Panx1 channels could trigger cellular swelling due to a boosted influx of H_2_O via aquaporins. At the other end, the massive release of gliotransmitters via astrocytic Cx43 hemichannels and Panx1 channels might reduce the viability and function of healthy neighboring neurons. Indeed, a substantial body of evidence indicates that these channels contribute to the development of multiple CNS disorders and diseases, including Alzheimer’s disease [119,161], epilepsy [162,163], meningitis [116], Parkinson’ disease [121], HIV-induced dementia [164,165], stroke [166,167], intracerebral hemorrhage [168], and spinal cord injury [117,169]. Solid lines depict fluxes of molecules through channels, whereas dashed lines indicate activation or induction.

**Figure 4 ijms-22-09503-f004:**
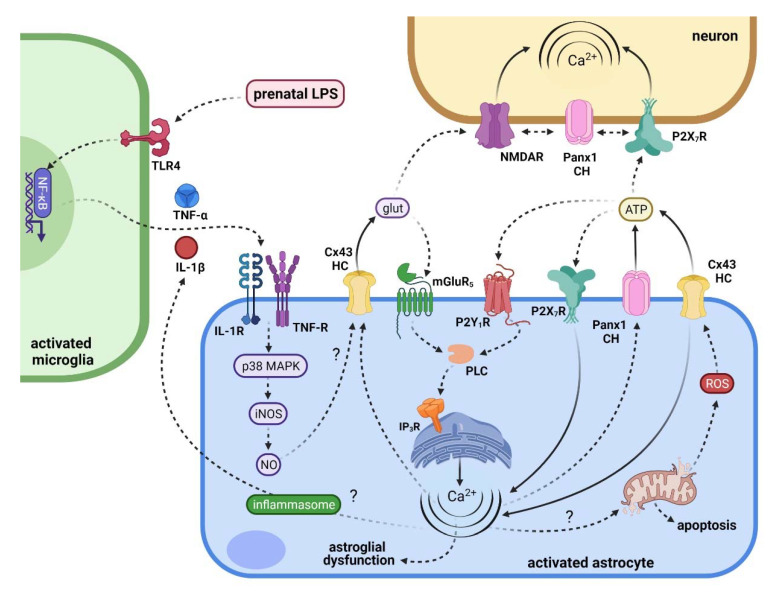
Diagram depicting the possible mechanisms and consequences of prenatal LPS-induced activation of Cx43 hemichannels/Panx1 channels in astrocytes. Indirectly, LPS exposure during pregnancy activates fetal and/or postnatal microglia, resulting in the TLR4 and NF-κB-dependent release of IL-1β and TNF-α. These cytokines, acting on their respective receptors, stimulate the p38MAPK/iNOS-dependent production of NO. The latter, possibly via de S-nitrosylation of Cx43, elicits the opening of Cx43 hemichannels and the subsequent release of glutamate and ATP through them. At one end, glutamate activates astrocyte mGluR5 receptors, causing the PLC-mediated stimulation of IP_3_ receptors and further release of Ca^2+^ stored in the endoplasmic reticulum. Meanwhile, ATP activates astrocyte P2X_7_Rs and P2Y_1_Rs, triggering the opening of Panx1 channels by increasing [Ca^2+^]_i_ and/or direct protein–protein interaction with P2X_7_Rs. Of note, alterations in [Ca^2+^]_i_ homeostasis mediated by Cx43 hemichannel or Panx1 channels might affect diverse aspects of astroglial function (e.g., morphology and pro-inflammatory profile). Cytosolic Ca^2+^ might trigger the collapse of mitochondrial membrane potential, causing oxidative stress (a well-known Cx43 hemichannel activator) and apoptosis via cytochrome C release through the mitochondrial transition pore and activation of caspase-3. Another possibility is that cytosolic Ca^2+^ could potentiate the release of pro-inflammatory cytokines (e.g., IL-1β) via the activation of the inflammasome. On the other hand, the excitotoxic release of glutamate and/or ATP through these channels could negatively impact neuronal [Ca^2+^]_i_ homeostasis and survival due to the activation of neuronal NMDARs, P2X_7_Rs, and Panx1 channels. Solid lines depict fluxes of molecules through channels, whereas dashed lines indicate activation or induction.

**Figure 5 ijms-22-09503-f005:**
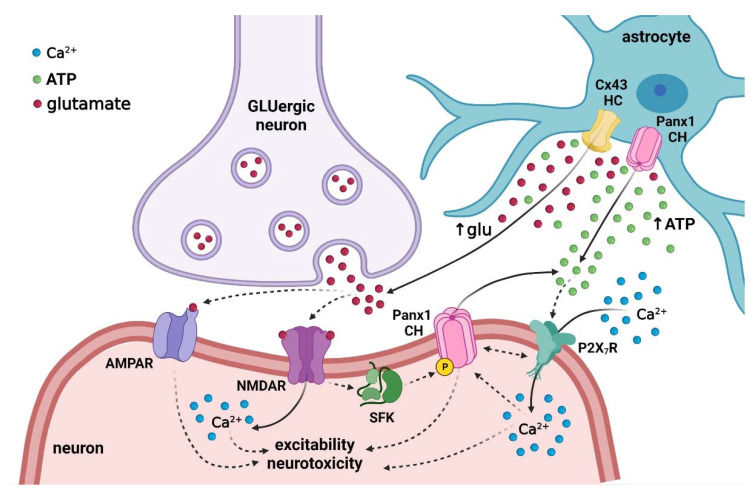
Possible detrimental roles on synaptic transmission of astroglial Cx43 hemichannels and Panx1 channels in prenatally LPS-exposed offspring. Prenatal LPS exposure raises postnatal cytokine brain levels, causing the opening of Cx43 hemichannels and Panx1 channels in astrocytes. The latter underpins the release of astrocytic ATP and glutamate towards the synaptic cleft. ATP could stimulate P2X_7_Rs, whereas glutamate might activate NMDARs and AMPARs in postsynaptic terminals of glutamatergic circuits. The downstream signaling of these receptors triggers the increase of [Ca^2+^]_i_, a phenomenon that the activation of neuronal Panx1 channels could exacerbate. This responsecould occur at least by two mechanisms: (i) via protein–protein interactions between Panx1 and P2X_7_Rs or (ii) due to the phosphorylation of Panx1 as a result of Src kinase (SFK) action mediated by the metabotropic function of NMDARs. The persistent activation of hemichannels/pannexons at the synaptic cleft might create a self-perpetuating loop of [Ca^2+^]_i_ imbalance with substantial and detrimental consequences for proper synaptic transmission and neuronal survival. Solid lines depict fluxes of molecules through channels, whereas dashed lines indicate activation or induction.

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
