# Peer review of "Astroglial Hemichannels and Pannexons: The Hidden Link between Maternal Inflammation and Neurological Disorders"

_ijms, 2021, doi:10.3390/ijms22179503_

Round 1
Reviewer 1 Report
Overall, I think this is a highly relevant and interesting review. Given that I am not an expert in the field, my impression that this is well-written and the citations appear to support the text. I like the abstract and conclusion. They draw in the reader and help conclude how this research relates to human development.
That said, there are several minor corrections that should be made before publishing, that I will mention below.
First, if the authors feel this is relevant, it would be interesting to have them include how their lab has contributed to the field. What were their results and how does it generally fit with the findings of other labs?
Next, regarding the timeline of fetal development, when during development does this signaling acting on Connexins and Pannexons occur? When do they start being expressed in the plasma membrane of brain cells of a human and rodent fetus? Does the timing of studies address this question? I’m not sure if it makes sense to have this in the intro or when connexions are introduced, but maybe that could be a reason for some discrepancies in the field.
Third, in some of the figures, there are both solid and dashed lines. Please define in the figure captions what do solid vs dashed indicate.
Some more specific comments throughout the paper:
122: there is a line break in the sentence (formatting issue)
166-168: which animal models does this refer to?
Could you define “maternal administration”? I.e. define if it is IP injection vs. I.v. (if that matters) oral administration to the female when pregnant and if the infection is assumed to be passed along to offspring. if it is injected into the fetus while in the mother, or if the mom has already given birth and so it is just passed through breast milk or breathing on them. How is it verified offspring have the infection?
188: accompanied of. Typo? Should it be “with”
241-243: Where does the calcium overload come from? Does this refer to Ca2+ overload of the astrocytes or the neuron.
Figure 3: There is a lot more detailed signaling going on in the figure than is described. Instead of repeating what is in the text, how about a better detailed description of what you are showing in the figure. For example: is the left and right both the same cell type? Why are they different colors and why do they have different number and appearance of cellular processions? Are the neurotransmitters going out into extracellular space in one but directly into the next cell in the other? Perhaps labeling of intracellular vs extracellular sections would also be helpful, and define if the extracellular is facing a gap junction, and which cell types are connected (astrocyte:astrocyte? Or glial cell:neuron). Is this intracellular side of the astrocyte?
Also, the signaling is shown in Figure 4 in better detail, so perhaps just leave out the signaling in Fig. 3 since it is discussed in the next section and Fig. 4.
Generally, how well do the animal model studies link to those disorders found in in human offspring?
278: typo? “while modulating?
Would the prenatal signaling process also be dependent on timing, in reference to whether a blood-brain barrier was already developed in the fetus?
Figure 4: The diagram shows the MAPK pathway completely separated from the Cx43 channels acted by Ca2+ signaling, but would the outcome of Glut/ATP release be similar? Or if not, is it because it is in a different cellular location? Would be helpful to have that explanation of why it is separated.
309: receptors being switched off by spread of ATP to distant areas…. Possibly recommend different wording such as: “..switch off due to the decrease of ATP by its diffusion from the micro domain within the cell, and its degradation by extracellular exonuclease”.
313: Appears to be a problem with the citation manager.
330: Because this is a question, I assume this is meant to start with “Is it” not “it is”.
348: typo suggestion…could significantly instead of sig. could.
349: line of though (t)
350: this is confusing because the administration of the blocker is prenatal, but then refers to adult hippocampus astrocytes. Could you elaborate on this discrepancy a bit? Also, how well-developed is the blood brain barrier (bbb) in the fetus at the time of administration? Or is this looking at the mother’s bbb?
359: typo? Although is?
The use of the word “tackled” is used throughout. While this is an interesting term to use, it would be useful to have a more descriptive term based on the context. For example, does it mean inhibited activity?, reduced expression? Reversed activity? Example in 375-377: does tackled mean that the cell death was it completely reversed? Indicate that here.
380: spelling: impaiment
393: suggestion: “results in neuronal toxicity” as defined by…which observations?
416 spelling error “suujectied”
417 spelling/ grammar error gliotrasnmiter” influence(s)
418 spelling error candidate
430: grammar suggestion: system being a “first line”…
I really like the last paragraph. I think this drives home the point that this inflammatory signaling pathway leading to prenatal damage either short or longterm, can be caused not just by infections but also driven by lifestyle or environmental exposures. If you are able to refer to other known cases of exposure activating hemichannels it would help support the importance and translatability of this type of research.
Author Response
Reviewer 1.
Comments and Suggestions for Authors
Overall, I think this is a highly relevant and interesting review. Given that I am not an expert in the field, my impression that this is well-written and the citations appear to support the text. I like the abstract and conclusion. They draw in the reader and help conclude how this research relates to human development.
That said, there are several minor corrections that should be made before publishing, that I will mention below.
First, if the authors feel this is relevant, it would be interesting to have them include how their lab has contributed to the field. What were their results and how does it generally fit with the findings of other labs?
Response: We thank the reviewer for this comment. Actually, an important part of what was described in sections 5 and 6 come from studies of our Laboratory (see PMID: 26096155; PMID: 31680871). With this in mind, we believe that it is unnecessary to deepen our contribution to the field throughout the manuscript.
Next, regarding the timeline of fetal development, when during development does this signaling acting on Connexins and Pannexons occur? When do they start being expressed in the plasma membrane of brain cells of a human and rodent fetus? Does the timing of studies address this question? I’m not sure if it makes sense to have this in the intro or when connexions are introduced, but maybe that could be a reason for some discrepancies in the field.
Response: Thank you very much for this comment.
Third, in some of the figures, there are both solid and dashed lines. Please define in the figure captions what do solid vs dashed indicate.
Response: We really appreciate this comment. In this new version of our manuscript, we included the information in the figure legends in order to address this suggestion. Solid lines depict fluxes of molecules through channels, whereas dashed lines indicate activation or induction. In this context, we amended solid and dashed lines depicted in figure 5 with the purpose of made them consistent with the above definition.
Some more specific comments throughout the paper:
122: there is a line break in the sentence (formatting issue)
Response: Thank you very much for this comment. This was amended according to the reviewer´s suggestion
166-168: which animal models does this refer to?
Could you define “maternal administration”? I.e. define if it is IP injection vs. I.v. (if that matters) oral administration to the female when pregnant and if the infection is assumed to be passed along to offspring. if it is injected into the fetus while in the mother, or if the mom has already given birth and so it is just passed through breast milk or breathing on them. How is it verified offspring have the infection?
Response: Either bacterial or viral infections during gestation have been modeled employing three major immunogenic approaches, particularly, administration of lipopolysaccharide (LPS), influenza virus or polyinosinic:polycytidylic acid (poly IC) to the pregnant rodent. LPS or poly IC have been injected intraperitoneally, intravenously or subcutaneously, while influenza virus is mostly administered intranasally. The evidence discussed in our review refer to studies in which immunogens have been administered systemically to pregnant animals. This is because studies in which LPS is administered directly to the fetus or locally into the uterus or cervix may differ substantially from systemic maternal LPS administration since LPS does not enter the fetal compartment when administered systemically to the pregnant dam (PMID: 7953189; PMID: 16189509).
The gestational age at which immunogens are usually administered varies across these studies on prenatal inflammation, ranging from daily administration of LPS or poly IC every day throughout pregnancy to a single administration on 1 day early or late in gestation. Despite the above, most studies use i.p. administration on pregnant mice during a specific window time in late gestation: embryonic day 17-18. In these models there is not infection neither in the mother nor the offspring. On the other hand, influenza virus infects and crosses the placenta, directly infecting the placental and fetal membranes, and leading to apoptosis and placentitis (PMID: 32911797). Offspring infection is usually verified by ELISA, PCR or western blotting against major envelope proteins of influenza virus.
In this new version of our manuscript, we included a special reference to the comprehensive review of Patricia Boksa (see third paragraphs of introduction section) (PMID: 20230889), who deeply discusses the pros and cons of different models of prenatal inflammation.
188: accompanied of. Typo? Should it be “with”
Response: Thank you very much for this comment. We amended this typo according to the reviewer´s suggestion
241-243: Where does the calcium overload come from? Does this refer to Ca2+ overload of the astrocytes or the neuron.
Response: Thank you very much for this comment. We rewrote and amended this paragraph with the purpose of address this comment. Hemichannels are permeable to Ca2+, which could allow its influx to the cytosol during pathological conditions. In the same line, Panx1 channels release ATP to the interstitial space, which activate purinergic receptors, causing the entry of extracellular Ca2+ or its release from intracellular stores. Both mechanisms could directly or indirectly cause calcium overload and this is a general mechanism of cell damage that may occur in different cell types, including astrocytes and neurons.
Figure 3: There is a lot more detailed signaling going on in the figure than is described. Instead of repeating what is in the text, how about a better detailed description of what you are showing in the figure. For example: is the left and right both the same cell type? Why are they different colors and why do they have different number and appearance of cellular processions? Are the neurotransmitters going out into extracellular space in one but directly into the next cell in the other? Perhaps labeling of intracellular vs extracellular sections would also be helpful, and define if the extracellular is facing a gap junction, and which cell types are connected (astrocyte:astrocyte? Or glial cell:neuron). Is this intracellular side of the astrocyte?
Response: Thank you very much for this comment. We redraw and amended this figure with the purpose of address this comment. The left panel represent a healthy astrocyte (in green) during physiological conditions, whereas the right panel shows a reactive astrocyte (in gray, showing cellular hypertrophy) in a pathophysiological scenario. This new version of the figure shows both intracellular and extracellular compartments labeled to avoid confusions. This figure aims to show the physiological and pathophysiological signaling of hemichannels/pannexons but not gap junction channels in astrocytes.
Also, the signaling is shown in Figure 4 in better detail, so perhaps just leave out the signaling in Fig. 3 since it is discussed in the next section and Fig. 4.
Response: Thank you very much for this comment. The major goal of Figure 3 is to introduce major mechanisms by which hemichannels/pannexons could induce cellular damage in astrocytes. In contrast, Figure 4 aims to give a detail of possible mechanisms and signaling pathways involved the opening of these channels in the context of prenatal inflammation.
Generally, how well do the animal model studies link to those disorders found in in human offspring?
Response: Thank you very much for this comment. Most of human evidence comes from epidemiological studies of prospective nature, in which a specific infectious pathogen or inflammatory marker in prenatal life is accessible to quantitative measurements. The major part of these markers fit well with inflammatory profiles observed in prenatally- LPS exposed offspring in murine models. Thus, the match of animal and clinical studies is likely a very powerful approach to link developmental immune abnormalities with the neuropsychiatric disease risk. Of course, due to ethical and technical reasons, however, human epidemiological research cannot directly determine causality for those associations and is often limited in its capacity to unveil the downstream cellular and molecular mechanisms altering normal brain development. Thus, experimental research in animals offers an exceptional chance to overcome these limitations.
278: typo? “while modulating?
Response: Thank you for noting this error. In this new version of our manuscript, we amended this mistake.
Would the prenatal signaling process also be dependent on timing, in reference to whether a blood-brain barrier was already developed in the fetus?
Response: Thank you very much for this comment. The reviewer’s assumption is correct. As mentioned in the last paragraph of section 3, the influence of maternal inflammation on brain abnormalities showed in the offspring depends on gestational time (see a comprehensive review PMID: 20230889). The BBB matures during fetal life and is well formed by birth (PMID: 8316332). Transport mechanisms may continue to develop in mammals born in a relatively immature state, including rats and mice, and become fully functional only in the peri- or post-natal period (PMID: 1410396). Thus, it is plausible to speculate that in addition to the placental barrier, the BBB, could be act as another factor limiting the spread of inflammatory signaling.
Figure 4: The diagram shows the MAPK pathway completely separated from the Cx43 channels acted by Ca2+ signaling, but would the outcome of Glut/ATP release be similar? Or if not, is it because it is in a different cellular location? Would be helpful to have that explanation of why it is separated.
Response: Thank you for noting this error. The reviewer’s assumption is correct. In this new version of our manuscript, we amended the Figure 4 linking both pathways to the same channels to avoid confusions.
309: receptors being switched off by spread of ATP to distant areas…. Possibly recommend different wording such as: “..switch off due to the decrease of ATP by its diffusion from the micro domain within the cell, and its degradation by extracellular exonuclease”.
Response: We thank the reviewer for this suggestion. In the new version of the manuscript, we amended this phrase according to the reviewer’s suggestion.
313: Appears to be a problem with the citation manager.
Response: Thank you for noting this error. In this new version of our manuscript, we amended this mistake.
330: Because this is a question, I assume this is meant to start with “Is it” not “it is”.
Response: Thank you for noting this error. In this new version of our manuscript, we amended this mistake.
348: typo suggestion…could significantly instead of sig. could.
Response: Thank you for noting this error. In this new version of our manuscript, we amended this mistake.
349: line of though (t)
Response: Thank you for noting this error. In this new version of our manuscript, we amended this mistake.
350: this is confusing because the administration of the blocker is prenatal, but then refers to adult hippocampus astrocytes. Could you elaborate on this discrepancy a bit? Also, how well-developed is the blood brain barrier (bbb) in the fetus at the time of administration? Or is this looking at the mother’s bbb?
Response: We thank the reviewer for this comment. The administration of the blocker takes place during postnatal time in the offspring. Indeed, Tat-gap19 is administrated via intraperitoneal (i.p.) injections beginning on PND 30, as has been previously described to be useful in acute and long-lasting administration in rodents. A second dose was given on PND45 followed by injections in PND 60, 75, 90, and 105. As mentioned in a previous comment, the BBB is mature up birth in rodents. In this new version of the manuscript, we mention that Tat-gap19 administration occurs at postnatal time to avoid confusions to the reader.
359: typo? Although is?
Response: Thank you for noting this error. In this new version of our manuscript, we amended this mistake.
The use of the word “tackled” is used throughout. While this is an interesting term to use, it would be useful to have a more descriptive term based on the context. For example, does it mean inhibited activity?, reduced expression? Reversed activity? Example in 375-377: does tackled mean that the cell death was it completely reversed? Indicate that here.
Response: We thank the reviewer for this suggestion. In this new version of the manuscript, we replace this word for others with a more descriptive nature. Nevertheless, it is important to have in mind that most hemichannel blockers are used to prevent rather to reverse negative outcomes.
380: spelling: impaiment
Response: Thank you for noting this error. In this new version of our manuscript, we amended this mistake.
393: suggestion: “results in neuronal toxicity” as defined by…which observations?
Response: Thank you very much for this comment. In this new version of the manuscript, we included the information required by the reviewer.
416 spelling error “suujectied”
Response: Thank you for noting this error. In this new version of our manuscript, we amended this mistake.
417 spelling/ grammar error gliotrasnmiter” influence(s)
Response: Thank you for noting this error. In this new version of our manuscript, we amended this mistake.
418 spelling error candidate
Response: Thank you for noting this error. In this new version of our manuscript, we amended this mistake.
430: grammar suggestion: system being a “first line”…
Response: Thank you very much for this comment. This was amended according to the reviewer´s suggestion
I really like the last paragraph. I think this drives home the point that this inflammatory signaling pathway leading to prenatal damage either short or longterm, can be caused not just by infections but also driven by lifestyle or environmental exposures. If you are able to refer to other known cases of exposure activating hemichannels it would help support the importance and translatability of this type of research.
Response: Thank you very much for this comment. Actually, the two references mentioned at the end of the conclusion section are the ones linking prenatal lifestyle or environmental exposure during pregnancy with the opening of hemichannels/pannexons in the brain offspring.
Reviewer 2 Report
In this review, Prieto-Villalobos et al. provide a comprehensive review about the inflammatory activation of astroglial hemichannels and pannexons during the maternal period on the development of neurological disorders in the offspring. In general, the authors addressed a contemporary topic of interest to general readers. The first half introduces us to maternal infection on abnormal brain development, followed by a description of the pathophysiological role of astrocytes in the brain. The latter half focuses on the activation of astroglial hemichannels and pannexons by inflammatory mediators and their involvement in neurological disorders. The potential mechanisms involved in the action of the channels are also mentioned. In general, the manuscript is well organized. The diagrams and cartoons used in the figures are especially well presented and helpful in understanding the text description.
As for the weakness of the manuscript, many sentences are too long. It makes it difficult to read, follow and understand. The conjunction words used are inappropriate in many locations. There are also grammatical errors and inaccurate descriptions. The manuscript needs to be polished to enhance the readability.
Concerning the content of the review: 1) the factors, other than inflammatory mediators, in the activation of these channels and their induction of neurological disorders, such as drugs, ROS and pollutants, should be mentioned. 2) the mechanisms involved can be illustrated in more details, such as the possible existence of channel-mediated the amplification of inflammation and oxidative stress, etc. 3) It would be nice to explain why a transient activation of these channels could have a long-lasting effect.
Author Response
Reviewer 2
Comments and Suggestions for Authors
In this review, Prieto-Villalobos et al. provide a comprehensive review about the inflammatory activation of astroglial hemichannels and pannexons during the maternal period on the development of neurological disorders in the offspring. In general, the authors addressed a contemporary topic of interest to general readers. The first half introduces us to maternal infection on abnormal brain development, followed by a description of the pathophysiological role of astrocytes in the brain. The latter half focuses on the activation of astroglial hemichannels and pannexons by inflammatory mediators and their involvement in neurological disorders. The potential mechanisms involved in the action of the channels are also mentioned. In general, the manuscript is well organized. The diagrams and cartoons used in the figures are especially well presented and helpful in understanding the text description.
As for the weakness of the manuscript, many sentences are too long. It makes it difficult to read, follow and understand. The conjunction words used are inappropriate in many locations. There are also grammatical errors and inaccurate descriptions. The manuscript needs to be polished to enhance the readability.
Response: We thank the reviewer for this comment. This new version of our manuscript was edited by a service for English Language Editing, which greatly improved our manuscript and reduced grammatical, spelling, and other common errors.
Concerning the content of the review:
1) the factors, other than inflammatory mediators, in the activation of these channels and their induction of neurological disorders, such as drugs, ROS and pollutants, should be mentioned.
Response: We thank the reviewer for this comment. In this new version of the manuscript, we included a phrase (last part of section 4) referring the different pathological stimuli that could activate astrocytic hemichannels/pannexons in the CNS, including drugs, ROS and pollutants.
2) the mechanisms involved can be illustrated in more details, such as the possible existence of channel-mediated the amplification of inflammation and oxidative stress, etc.
Response: We thank the reviewer for this comment. We redraw and amended the Figure 4 with the purpose of address this comment. Particularly, we included new draws and text in the figure legend indicating possible pathways by which hemichannels/pannexons could serve as amplifiers of inflammation and oxidative stress.
3) It would be nice to explain why a transient activation of these channels could have a long-lasting effect.
Response: We thank the reviewer for this comment. In the new version of our manuscript, we included a text the second paragraphs of section 6 explaining how a transient activation of these channels could have a long-lasting effect in astrocytes.